# *Lactobacillus paracasei* HY7015 and *Lycopus lucidus* Turcz. Extract Promotes Human Dermal Papilla Cell Cytoprotective Effect and Hair Regrowth Rate in C57BL/6 Mice

**DOI:** 10.3390/molecules27238235

**Published:** 2022-11-25

**Authors:** Hayera Lee, Hyeonji Kim, Ji-Hyun Kim, Soo-Dong Park, Jae-Jung Shim, Jeong-Lyoul Lee

**Affiliations:** R&BD Center, hy Co., Ltd., 22, Giheungdanji-ro 24beon-gil, Giheung-gu, Yongin-si 17086, Republic of Korea

**Keywords:** *Lactobacillus paracasei*, *Lycopus lucidus* Turcz., cytoprotective effects, hair regrowth

## Abstract

Hair loss is a disease that requires accurate diagnosis and type-specific medical treatment. Many hair loss treatments have some side effects, such as hormone-related effects, so there is a need for safe and effective hair loss treatment. In this study, we investigated the effects of *Lactobacillus paracasei* HY7015 (HY7015) and *Lycopus lucidus* Turcz. (LT) extract on hair regrowth and protection. In vitro experiments were conducted to assess the effects of HY7015 and/or LT extract on human follicle dermal papilla cells (HFDPC) of cytoprotective functions such as proliferations, antioxidants, anti-inflammatory, and growth factor expressions. In animal experiments, we investigated hair regrowth rate, hair follicle formation and secretion of growth factors in telogenic C57BL/6 mice. We confirmed the cytoprotective effects of HY7015 and LT through regulations of proliferation, SOD and IL-1β in HFDPC. In mouse experiments, oral administration of HY7015 and LT promoted hair regrowth as well as hair follicle maturation in the dermal skin of C57BL/6 mice, and upregulated VEGF and IGF-1 growth factor levels in mouse serum. In summary, our data demonstrate that ingestions of HY7015 and LT can promote hair regrowth by enhancing cytoprotective effects and expressions of growth factors.

## 1. Introduction

Hair plays an important role in creating a physical barrier against the external environment to protect the scalp [1]. Hair loss refers to a state in which there is no hair in an area where hair is usually present, and generally refers to the loss of the thick hair of the scalp; unlike soft hairs, which are colorless and thin, the loss of scalp hair can cause not only cosmetic problems but also psychological stress. Although a direct relationship has not been revealed, recent studies have been conducted that show a statistical correlation between hair loss and prostatic hyperplasia, hypertension and metabolic syndrome [2,3,4]. Currently, drugs approved by the U.S. Food and Drug Administration (FDA) as being able to promote and mimic hair growth include minoxidil and finasteride [5]; however, these drugs have side effects, including systemic hair growth and decreased sexual function. Therefore, there is a need to develop methods to prevent hair loss and promote hair growth that are physiologically safe for humans, through promoting cytoprotective effects and growth factors in dermal papilla cells.

Probiotics are viable bacteria that improve the intestinal microbial balance and have beneficial effects on the health of the host [6,7]. *Lactobacillus* species can improve intestinal health and intestinal inflammation, and have various other functions, including as antioxidants, immunity enhancements and anti-obesity measures [8,9,10]. In addition, *Lactobacillus* species also affect hair growth: *Lactobacillus plantarum* expresses vascular endothelial growth factor (VEGF) and keratinocyte growth factor (KGF), which contribute to hair growth in vitro and in vivo [11,12,13]. Additionally, *Lactobacillus paracasei* HY7015 (HY7015), which investigates in this study, promotes secretion of hair growth factors in hair follicle dermal papilla cells and telogenic C57BL/6 mice [14].

*Lycopus lucidus* Turcz. (LT) is an herb belonging to the family Lamiaceae that has been used for centuries as a traditional Oriental medicine (Figure 1). Recent studies have shown that LT has various functionalities, including as treatment for allergies, diabetic retinopathy, and non-alcoholic fatty liver disease [15,16,17,18,19,20]. LT includes phenolic compounds, such as rosmarinic acid, ferulic acid, caffeic acid, and flavonoids, like luteolin, rutin, and quercetin [15,16]. The aerial parts of LT are used as a traditional phytomedicine to relieve thyroid dysfunction, as well as sedative, anti-inflammatory, wound-healing, promoting blood circulation and pain reliving agents; they are also used as a tonic in East Asia [17,18,21,22].

We hypothesize that LT, which is effective in blood circulation, wound healing, and anti-inflammation, would also be effective in hair growth. Furthermore, plant extracts show to enhance the viability and health effects of probiotic *Lactobacillus* strains [23,24,25]; therefore, we investigate the effects of HY7015+LT proliferation and protection against oxidative stress, also hair growth factors on HFDPC. Furthermore, we researched the effects of treatment with both HY7015 and LT together on hair regrowth, and evaluated the combined activity of these substances in telogenic mice.

## 2. Results

### 2.1. Cytoprotective Effect of HY7015 and/or LT Extract in HFDPC

#### 2.1.1. Effect of HY7015 and/or LT Extract on HFDPC Proliferations

To determine the effects of HY7015 and LT extract on hair growth, we assess the proliferation of HFDPC treated with these reagents. Proliferation of HFDPC cells treated with the positive control, MYE, was significantly higher (147.85 ± 0.36%, *p* < 0.05) than untreated controls. Additionally, treatment of HY7015 (135.13 ± 0.29%) and LT (148.54 ± 0.32%, *p* < 0.05) raised relative to untreated controls. Proliferation of HY7015+LT was mostly increased (158.95 ± 0.54%, *p* < 0.05) (Figure 2A). Hence, both HY7015 and LT extract increased dermal papilla proliferation, and cells in the HY7015+LT-treated group proliferated more strongly than those treated with each reagent separately.

#### 2.1.2. Antioxidants and Anti-Inflammatory Effects in HFDPC

To understand whether HY7015 and/or LT has the effects of antioxidants and anti-inflammatory, SOD1 activity and IL-1β proteins were investigated. The value for the control group would be set as one, with expression in the other groups calculated relative to the control. As shown in Figure 2B, SOD1 activity decreased by H_2_O_2_ only (38.98 ± 13.27%) and was recovered by HY7015+LT (78.28 ± 19.59%). There was a tendency to increase to 52.63 ± 7.71% with HY7015 treatment, but it was not significant. IL-1β protein levels were 21.19 ± 0.58, 54.96 ± 1.94, 52.57 ± 3.41, 70.21 ± 13.14, 50.55 ± 1.65, and 46.96 ± 2.94 pg/mL in control, H_2_O_2_ only, MYE, HY7015, LT, and HY7015+LT groups (Figure 2C). LT and HY7015+LT groups alleviated IL-1β protein levels in HFDPC.

### 2.2. HY7015 and/or LT Affected on VEGF Secretion by HFDPC and NIH3T3 Fibroblasts

We investigated the effects of HY7015 and/or LT influenced VEGF protein secretions by HFDPC in culture supernatants after treatment for 24 h. VEGF protein levels were 77.38 ± 8.56, 142.36 ± 2.96, 253.00 ± 73.78, 193.37 ± 28.99, and 290.96 ± 29.32 pg/mL in Control, MYE, HY7015, LT, and HY7015+LT groups in HFDPC (Figure 3A). It significantly increased in the HY7015 (253.00 ± 73.78 pg/mL, *p* < 0.001), LT (193.37 ± 28.99 pg/mL, *p* < 0.001) and HY7015+LT (290.96 ± 29.32 pg/mL, *p* < 0.001) groups. We also confirmed mRNA expression levels of VEGF by HFDPC after treatment for 24 h. The value for the control group was set as one, with expression in the other groups calculated relative to the control. Expression mRNA levels of VEGF were 2.74-fold, 1.07-fold, 1.20-fold, 3.25-fold in HFDPC treated with MYE, HY7015, LT, and HY7015+LT (Figure 3B). There was no significant difference between HY7015 and LT group; however, the mRNA levels of VEGF was significantly increased in HY7015+LT group (3.25-fold, *p* < 0.05).

Next, we investigated whether treatment with HY7015 and LT for 24 h affected VEGF protein and mRNA expression of growth factors in NIH3T3 fibroblasts prior to animal experiments. VEGF protein levels were 105.06 ± 0.17, 118.61 ± 1.56, 100.04 ± 8.59, 106.89 ± 1.65, and 116.44 ± 0.96 pg/mL in Control, MYE, HY7015, LT, and HY7015+LT groups (Figure 3C). VEGF secretions significantly increased in HY7015+LT group (*p* < 0.001). As shown in Figure 3D, mRNA expression levels of Vegf were 1.37-fold, 1.58-fold, and 1.55-fold in NIH3T3 fibroblasts treated with MYE, HY7015, and LT, respectively; these levels were slightly greater than those in the control group (1.02-fold), but the difference was not significant. However, in the HY7015+LT group, Vegf levels were significantly increased relative to untreated controls (1.78-fold, *p* < 0.05).

### 2.3. Effect of HY7015 and/or LT on Hair Regrowth in C57BL/6 Mice

We used a telogenic mouse model to examine the effects of oral ingestion of HY7015 and LT extract on hair regrowth. In our experiment, we assessed whether oral treatment with HY7015 and LT could promote hair growth for 5 weeks (Figure 4A). To observe the rate of hair regrowth, hair was removed from the same area on the backs of the mice before feeding commenced.

As shown in Figure 4B, the degree of hair regrowth was observed in hair removal areas. Hair regrowth rate in the positive control group (75.25 ± 4.21%, *p* < 0.001) was higher than that in the control group (9.48 ± 4.66%). Further, rates of hair regrowth were increased by 62.20 ± 5.98% (*p* < 0.001), 67.82 ± 7.76% (*p* < 0.001), and 74.98 ± 7.23% (*p* < 0.001) in the HY7015, LT, and HY7015+LT groups, respectively. The HY7015 and LT groups had less hair regrowth than the positive control (MYE) group, whereas levels of growth in the HY7015+LT group were similar to those in the MYE group.

In addition, we compared the hair regrowth in mice every week for 5 weeks between groups (Figure 4C). There was no significant difference in the degree of hair growth between all groups after feeding for 3 weeks. In the control group, hair regrowth rates were 3.12 ± 1.09% and 9.48 ± 4.66% in the 4th and 5th weeks, respectively. Hair regrowth rates in the MYE group were 41.20 ± 7.30% (*p* < 0.001) and 75.25 ± 4.21% (*p* < 0.001) in the 4th and 5th weeks, respectively, relative to the control group. In the HY7015 group, growth rates were 35.38 ± 7.69% (*p* < 0.01) and 62.20 ± 5.98% (*p* < 0.001) in the 4th and 5th weeks, respectively. LT extract led to similar hair regrowth to that observed in the HY7015 group up to the 4th week of ingestion, with rates of 34.55 ± 11.03% (*p* < 0.05) and 67.82 ± 7.76% *(p* < 0.001) in the 4th and 5th weeks, respectively. Finally, hair regrowth in the HY7015+LT-treated group showed a similar trend to that in the MYE-treated group over the 5 weeks, with rates of 40.49 ± 10.80% (*p* < 0.01) and 74.98 ± 7.23% (*p* < 0.001) in the 4th and 5th weeks, respectively.

### 2.4. Effect of HY7015 and/or LT on Hair Follicles and Dermal Skin Thickness in C57BL/6 Mice

Hair regrowth occurs in association with increases in the number of hair follicles and dermal skin thickness. Therefore, by measuring numbers of hair follicles and dermis layer thickness, the effects on hair regrowth can be predicted [26]. To analyze the number and shape of hair follicles, we examined longitudinal and transverse sections of mouse dorsal skin tissue samples by H&E staining (Figure 5A,B, Appendix A). As shown in Figure 5C, the mean number of hair follicles in the MYE group (167.29 ± 12.00 × 10^4^ hair follicles/μm^2^, *p* < 0.001) was significantly higher than that in the control group (4.13 ± 2.45 × 10^4^ hair follicles/μm^2^), as were those in the HY7015, LT, and HY7015+LT groups, at 160.57 ± 7.97 × 10^4^ (*p* < 0.01), 158.71 ± 7.86 × 10^4^ (*p* < 0.01), and 163.80 ± 19.27 × 10^4^ (*p* < 0.001) hair follicles/μm^2^, respectively; these numbers were slightly lower than those in the positive control group (MYE) but indicate a similar follicle-generating effect.

Measurement of dermis layer thickness was performed using ImageJ software. As shown in Figure 5D, similar dermis skin thickness developed in MYE, HY7015, LT, and HY7015+LT-treated groups, with increases of 65.02 ± 20.97 (*p* < 0.001), 58.51 ± 18.46 (*p* < 0.001), 54.44 ± 15.81 (*p* < 0.001), and 63.72 ± 19.33 (*p* < 0.001) μm, respectively, compared with the untreated group (21.75 ± 7.96 μm). Skin thickness in the HY7015+LT group was comparable to that in the positive control (MYE) group.

These results indicate that administration of HY7015 and LT extracts combined affects hair regrowth by upregulating the number of hair follicles and dermal layer thickness, similarly to the positive control.

### 2.5. Effect of HY7015 and/or LT on Growth Factor Levels in Mouse Serum

Levels of the growth factors, VEGF and IGF-1, which contribute to hair growth were analyzed by ELISA in serum samples separated from mouse blood. As shown in Figure 6A, VEGF secretion levels in treated groups, including MYE (157.15 ± 7.56 pg/mL, *p* < 0.001), HY7015 (157.35 ± 5.79 pg/mL, *p* < 0.001), LT (155.70 ± 1.94 pg/mL, *p* < 0.001), and HY7015+LT (159.34 ± 1.97 pg/mL, *p* < 0.001) were similar, and were significantly higher than those in the untreated group (117.24 ± 3.16 pg/mL). Further, serum levels of IGF-1 were significantly increased in the MYE, HY7015, LT, and HY7015+LT groups, at 533.13 ± 12.31 (*p* < 0.05), 560.80 ± 14.58 (*p* < 0.01), 558.67 ± 10.32 (*p* < 0.01), and 601.52 ± 17.2 (*p* < 0.001) ng/mL, respectively (Figure 6B). Levels of both VEGF and IGF-1 were highest in the HY7015+LT group, suggesting that HY7015+LT ingestion can contribute to hair regrowth by promoting growth factor secretion in mice.

### 2.6. Rosmarinic Acid Content of LT Extract

LT has various phenolic compounds such as rosmarinic acid, ferulic acid, and caffeic acid, and flavonoids such as luteolin, rutin, and quercetin [15,16]. We identified rosmarinic acid content, which is known to be related to hair protection and growth [27,28,29,30]. Rosmarinic acid in the LT extract were quantified using HPLC, and rosmarinic acid content of LT extract used in this study were 67.64 μg/mL (Table 1, Appendix A).

### 2.7. Rosmarinic Acid Affected on VEGF Expression by HFDPC

To confirm whether rosmarinic acid (RA) affected VEGF mRNA expressions in HFDPC, rosmarinic acid was treated for 24 h. mRNA levels of VEGF were 3.89-fold (*p* < 0.001), 3.95-fold (*p* < 0.001), and 1.47-fold (*p* < 0.05) in HFDPC treated with MYE, 5 μg/mL and 50 μg/mL of RA (Figure 7).

## 3. Discussion

Human hair growth is necessary to conserve body temperature and protect from the external environment. As hair loss reduces quality of life, many studies to address this problem have been conducted [31]. Minoxidil and finasteride are currently approved by the U.S. FDA for preventing hair loss; however, these two drugs have various side effects [32,33,34,35]. Therefore, it is necessary to identify safe substitutes that promote hair growth and prevent hair loss in humans. In a previous study, we confirmed that the probiotic HY7015 can promote hair growth through angiogenesis supplying essential nutrients around cells in the blood flow [14]. In this study, we tested another natural product that promotes hair growth in a similar way to HY7015 and found that it can act together with probiotics to improve hair growth. Measurements of the proliferation of cells involved in hair growth and the elevation of angiogenesis-related factors were conducted in this study to confirm the efficacy of these products in promoting hair growth.

Since dermal papilla cells are the core of hair growth, it can be predicted that their proliferation will promote hair growth [36]. Hence, HFDPCs’ proliferation can be predicted to promote hair follicle formation and hair growth. In this study, we found that HY7015+LT can improve HFDPC proliferations (Figure 2A). HY7015 and LT also showed a tendency to increase proliferation. It was confirmed that proliferation was greatly increased due to the synergistic effect of the HY7015 and LT.

Oxidative stress can lead to hair loss by causing cell death of hair follicle, inducing hair damage such as thinning of hair thickness [37,38]. Therefore, it can help hair health by suppressing or eliminating the generation of excessive active oxygen through antioxidant activity and reducing oxidative stress. In particular, superoxide dismutase plays an important role in regulating the formation of reactive oxygen species. Imbalances in the immune system are also known to lead immune cells to attack hair follicles, causing inflammation, damage to the hair, and hair loss. Inflammatory cytokines such as IL-1 are known to induce cell death of hair cells and hair loss. As a result, SOD1 activity decreased by H_2_O_2_ was significantly recovered by HY7015+LT. In addition, IL-1β, which was increased by H_2_O_2_, was significantly decreased by LT and HY7015+LT. These results indicated that treatment with HY7015 and LT together has a stronger effect on SOD1 activity and protein expressions of IL-1β than either individual treatment (Figure 2B,C).

Furthermore, VEGF factors play critical roles in promoting hair growth and regulating the transition from anagen to catagen during the hair growth cycle. VEGF contributes to follicle formation during hair growth, as well as regulating physiological and pathological angiogenesis. It is known that VEGF strengthens hair capillaries and creates new blood vessels to supply nutrients to the hair and improve its condition. It has also been studied that Minoxidil upregulates VEGF in HFDPC [39,40].

We preferentially confirmed the VEGF proteins and mRNA expressions by HY7015 and LT in HFDPC. As a result, proteins and mRNA expressions of VEGF were strongly increased when the HY7015+LT was treated with HFDPC for 24 h (Figure 3A,B). The NIH3T3 cell line is derived from mouse embryonic stem cells that secrete several growth factors, which stimulate keratinocyte growth. Keratinocyte growth can regulate differentiation of epithelial tissues and stem cells in the hair follicle [41,42,43]. Therefore, we identified VEGF proteins and Vegf mRNA levels to investigate the effect of HY7015 and LT on growth factors in NIH3T3 fibroblasts prior to animal experiments. Similarly, VEGF levels were significantly elevated in the HY7015+LT group (Figure 3C,D). These results indicated that treatment with HY7015 and LT together has a stronger effect on VEGF expressions, one of the important factors for hair growth, than either individual treatment.

The effects of HY7015 and LT extract ingestion on hair regrowth were evaluated using 7-week-old C57BL/6 mice entering telogen stage. C57BL/6 mice are a suitable model for screening substances that promote hair regrowth because the pigment produced by follicle melanocytes affects their truncal pigmentation only during anagen [44,45,46]. After the 4th week, there was a significant increase in hair and regrowth rate relative to the untreated group. Treatment with either HY7015 or LT alone caused significant increases in regrowth, while weekly hair regrowth rates in the group treated with HY7015+LT increased to a similar degree to that observed in mice treated with MYE (Figure 4C). Hence, ingestion of HY7015 and LT induced rapid hair growth relative to the control group.

The number and size of hair follicles increased during progression to the anagen hair growth phase. As the number and proportion of hair follicles rises, the thickness of the epidermal skin layer also increases [26]. H&E of transverse and longitudinal skin sections demonstrated that HY7015 and LT directly influence hair follicle development in mouse skin tissue. Transverse sections were used to measure hair follicle numbers and longitudinal sections to analyze dorsal skin thickness and hair growth cycle progression. The proportions of hair follicles in the MYE, HY7015, and/or LT-treated groups were all higher than those in mice fed the control diet, indicating that intake of those substances affects follicle proportions in anagen phase. Dorsal skin thickness measurement revealed a similar tendency to follicle numbers, with all groups treated with substances having significantly increased thickness relative to the untreated group (Figure 5). In particular, ingestion of HY7015 and LT extract increases the number of hair follicles and stimulates hair growth cycle progressions.

VEGF levels of mice serum were significantly higher in the HY7015 and LT extract-treated groups than in the control group, and were similar in all groups, except untreated controls (Figure 6A). HY7015 and LT intake stimulated IGF-1 secretion even more strongly than the positive control. Further, administration of HY7015+LT was associated with the highest levels of IGF-1 secretion among all groups (Figure 6B). These data show that HY7015 and LT extract can promote growth factor secretion in vivo.

Mixing probiotics with natural products increases the survival rate of lactic acid bacteria [23,24,25], and probiotics are known to affect absorption and metabolism of rosmarinic acid in the gastrointestinal tract model [47]. LT is known to have active ingredients such as phenolic compounds and flavonoids [15,16]. Previous studies have shown that rosmarinic acid has phenolic compounds that exhibit biological activities such as anti-inflammatory, antioxidant and cytoprotective effects [48,49,50,51]. In addition, rosmarinic acid alleviates cellular damage caused by oxidative stress induced by UVB in human keratinocytes [52]. Therefore, the content of rosmarinic acid, which is known to be the most abundant in LT [53], was measured. Our data show that the rosmarinic acid content of the LT water extract was 67.64 μg/mL (Table 1, Appendix A). LT containing rosmarinic acid has protective effects on hair cells, which may affect hair regrowth. Therefore, we evaluated the effect of rosmarinic acid on VEGF mRNA expression in HFDPC. VEGF expression in HFDPC was significantly increased at low concentration of rosmarinic acid rather than high concentration (Figure 7). It is necessary to explore the optimal concentration of rosmarinic acid that affects hair health.

Recently, studies have reported the relations between intestinal environment and skin health [54,55]. Gut microbial metabolites such as short chain fatty acids (SCFAs), dopamine and serotonin are known to directly or indirectly affect skin health through anti-inflammation, hair follicle inhibition, and melatonin modulation. Furthermore, genes associated with alopecia areata affect intestinal colonization of microorganisms that induce Th1 responses, which can lead to abnormal growth of hair follicle cells. In addition, the benefits of Lactobacillus genus consumption have also been studied on skin health through gut-modulating and immune responses [56,57]. HY7015 is considered to have the effects on genes related to hair growth by regulating intestinal balance and immune response system. However, further research, including clinical studies, should be conducted to support this. In addition, diverse factors influence the growth and inhibition of microorganisms. In particular, probiotics are known to influence growth during fermentation by carbon, nitrogen, trace elements as well as polyphenols [58]. Additionally, recent studies have shown that fermentation for 24 h with polyphenols from various plants regulates the growth and inhibition of microorganisms [59]. Various polyphenolic compounds related to LT have been reported, but research on the relationship between LT and probiotics has not progressed. Therefore, our paper provides a potential possibility that LT affects HY7015 growth.

Further studies are needed to clarify the mechanisms by which these substances influence hair regrowth by identifying which function(s) of HY7015 and component(s) of LT influence cell proliferation, hair regrowth-related gene expression, and hair cycle progression.

## 4. Materials and Methods

### 4.1. Sample Preparation

In total, 100 g of dried natural plant and aerial LT samples (Humanherb, Daegu-si, Republic of Korea) was ground, mixed, and stirred for 6–8 h at 95–98 °C in 1 L distilled water. Extract was concentrated under reduced pressure using a rotary vacuum concentrator (EYELA, Tokyo, Japan), and the concentrated extract was freeze-dried at –80 °C (Operon, Gimpo-si, Republic of Korea) and then powdered. LT extract was stored at –20 °C until use.

*Lactobacillus paracasei* HY7015 was isolated from Makgeolli, a traditional Korean fermented liquor made from rice, and maintained as frozen stock in MRS broth (BD Difco, Sparks, MD, USA) containing 20% (*v*/*v*) glycerol at –80 °C. HY7015 was incubated in MRS broth at 37 °C for 18 h and then harvested by centrifugation (4000× *g*, 20 min). Cell pellets were washed and resuspended in sterile phosphate-buffered saline (PBS) for in vitro assays. Cells were stored at –20 °C until use. For in vivo assay, fresh cultured HY7015 was prepared by freeze-drying and added to feed. Strains were identified through 16S rRNA sequencing (Macrogen, Inc., Seoul, Republic of Korea) using universal rRNA gene primers (785F and 907R). 16S rRNA sequencing results were compared to the Genbank database via the Basic Local Alignment Search Tool (BLAST) of the National Center for Biotechnology Institute (NCBI). The sequenced strain was named *L. paracasei* HY7015 and deposited with KCTC (KCTC, Jeonge-up, Jeollabuk-do, Republic of Korea) with accession number KCTC14004BP.

Minoxyl-S (Hyundai Pharm. Co., Ltd., Seoul, Republic of Korea) was purchased and used as positive control. The main component of Minoxyl-S is medicinal yeast extract (MYE), which supplies essential nutrients for hair growth.

### 4.2. Cell Culture and Treatments

Human hair follicle dermal papilla cells (HFDPC) were purchased from Cell Applications (San Diego, CA, USA). Cells were maintained in HFDPC medium and supplement kit (Cell Applications) in T-75 flasks at 37 °C in a humidified 5% CO_2_ atmosphere. Cells were seeded into 12-well plates at 4 × 10^4^ cells/well. Attached cells were washed with phosphate-buffered saline, treated by adding Dulbecco’s modified Eagle’s medium (DMEM) with DMSO only (Control) or medicinal yeast extract (MYE, 50 µg/mL; positive control), HY7015 (1 × 10^6^ CFU/mL), natural plant extracts (50 µg/mL), for 24 h.

Mouse embryonic NIH3T3 fibroblasts were purchased from Cell Applications (San Diego, CA, USA). Cells were maintained in Dulbecco’s modified Eagle’s medium (DMEM) supplement with 10% heat-inactivated bovine calf serum (BCS) and 1% penicillin-streptomycin (P/S) at 37 °C in a humidified 5% CO_2_ atmosphere. Cryopreserved cells were cultured in DMEM with 10% BCS and 1% P/S in T-75 flasks for 24 h to reach 80% confluence. Cells were seeded into 24-well plates at 1 × 10^5^ cells/well, maintained in DMEM containing 2% BCS and untreated (Control) or medicinal yeast extract (MYE, 50 μg/mL; positive control), HY7015 (1 × 10^6^ CFU/mL), natural plant extracts (50 μg/mL), for 24 h.

### 4.3. Proliferation Analysis

HY7015 (1 × 10^6^ colony-forming units (CFU)/mL), LT extract (50 µg/mL), a HY7015+LT (HY7015, 1 × 10^6^ CFU/mL; LT, 50 µg/mL) were treated for 24 h. After treatment, HFDPC proliferation were evaluated using the cell counting kit-8 (CCK-8) assay (Dojindo Molecular Technologies, Inc., Kumamoto, Japan). CCK-8 solution (10 µL/well) was added to each well of the plate and incubated for 4 h, 37 °C in a CO_2_ incubator, and then absorbance was measured at 450 nm (BioTek^®^ Synergy HT, Santa Clara, CA, USA).

### 4.4. Enzyme-Linked Immunosorbent Assay (ELISA) In Vitro

To determine superoxide dismutase 1 (SOD1), IL-1β, and VEGF contents in HFDPC, the ab119520 -Superoxide Dismutase 1 Human ELISA kit (Abcam, Trumpington, Cambridge, UK), Human IL-1β ELISA Set (BD Biosciences, Franklin Lakes, NJ, USA), and Quantikine ELISA human VEGF (DVE00) kits (R&D Systems, Minneapolis, MN, USA) were used according to the manufacturer’s instructions. For H_2_O_2_ treatment the samples were treated for 20 h and then treated with 1000 μM H_2_O_2_ for 4 h, and then the supernatant was collected, and protein was measured. To determine VEGF contents in NIH3T3 fibroblasts, the Mouse VEGF (MMV00) Quantikine ELISA kits (R&D Systems, Minneapolis, MN, USA) were used according to the manufacturer’s instructions. NIH3T3 cells treated with samples for 24 h, and then the supernatant was collected, and protein was measured.

### 4.5. Total RNA Extraction, cDNA Synthesis, and Quantitative Real-Time PCR Analysis

HFDPC and NIH3T3 cells treated with HY7015, LT, HY7015+LT, MYE and untreated group (Control) for 24 h were washed twice in PBS, and then total RNA was extracted from washed cells using the easy-spin Total RNA Extraction Kit (iNtRON Biotechnology, Seoul, Republic of Korea). Extracted total RNA samples were quantified using NanoDrop 2000 (Thermo Scientific, Waltham, MA, USA) and stored at –20 °C until use for gene expression analysis. Total RNA (2 µg aliquots) was reverse-transcribed into cDNA using an Omniscript Reverse Transcription Kit (Qiagen, Hilden, Germany), with dNTP mix, 10× Buffer RT, Omniscript reverse transcriptase, and RNase free water, and then incubated at 37 °C for 1 h according to the manufacturer’s instructions. cDNA samples were amplified using the QuantStudio 6-Flex Real-time PCR System (Applied Biosystems, Foster City, CA, USA) and Gene Expression Master Mix (Applied Biosystems). Real-time PCR (RT-PCR) was conducted using mouse-specific TaqMan Gene Expression Assays as follows: vascular endothelial growth factor A (VEGFA; Hs_00173626_m1, *Vegf*; Mm00437306_m1). Expression data were normalized using those of the housekeeping gene glyceraldehyde-3-phosphate dehydrogenase (GAPDH; Hs99999905_m1, *Gapdh*; Mm99999915_g1).

### 4.6. Animal Experiments

Animal experimental procedures were approved by the Ethics Review Committee of R&BD Center, hy Co., Ltd., Korea (AEC-2021-00006-Y). Female C57BL/6 mice, aged six weeks, were purchased from KOATECH (Pyeongtaek-si, Gyeonggi-do, Republic of Korea). All mice were bred at constant temperature (22 ± 1 °C) and humidity (55 ± 10%) with a 12 h light/12 h dark cycle. After 1 week of acclimatization, mice were divided into five groups (*n* = 8 per group) fed with normal control diet or supplemented as follows: 200 mg/kg MYE, HY7015 (1 × 10^8^ CFU/day), 100 mg/kg LT extract (LT), and HY7015 (1 × 10^8^ CFU/day) with 100 mg/kg LT extract (HY7015+LT). After 5 weeks, mice were sacrificed, and blood samples were extracted (Figure 8). Blood was allowed to stand at room temperature for 30 min and then centrifuged at 3000 *g* for 20 min to separate serum. Separated serum and skin tissue samples were stored at –80 °C until use.

### 4.7. Histological Analysis of Dorsal Skin Tissue

Dorsal skin tissue samples from each mouse were fixed in 10% formalin solution, and then sectioned and stained with hematoxylin and eosin (H&E) on slides by T&P Bio (Gwangju-si, Republic of Korea). Dorsal skin thickness was measured under a Zeiss Axiovert 200M microscope (Carl Zeiss AG, Thornwood, NY, USA) at 100× magnification. Images of dorsal skin tissue were analyzed using ImageJ software (National Institutes of Health, Bethesda, MD, USA) to measure the number of hair follicles per unit area (×10^4^/µm^2^).

### 4.8. Analysis of Growth Factors in Mouse Serum

To analyze growth factor concentrations in mouse serum, separated serum samples were analyzed using Mouse VEGF (MMV00) and Mouse/Rat IGF-I/IGF-1 (MG100) Quantikine enzyme-linked immunosorbent assay (ELISA) kits (R&D Systems, Minneapolis, MN, USA), according to the manufacturer’s instructions.

### 4.9. Measurement of Rosmarinic Acid Content

The content of rosmarinic acid was measured by HPLC-DAD referring to the previous methods [60,61]. Approximately 0.5 g of each sample was extracted by 10 mL of methanol-H_2_O (7:3), followed by ultrasonication at room temperature for 30 min. After centrifugation, it was transferred to a 25 mL volumetric flask and made up to volume with 70% methanol. Then, it was filtered through 0.22 μm syringe filter and analyzed. Rosmarinic acid (Sigma-Aldrich, St. Louis, MO, USA) was used as the standard. 330 nm was selected as the wavelength for UV detection. A CAPCELL PAK C18 (OSAKA SODA, UG120, 4.6 mm I.D*250 mm, 5 μm) column was used and the injection volume was 10 μL. Elution was performed at a flow rate of 1.0-mL/min at 25 °C. Two mobile phases A and B were used. Mobile phase A was 0.1% (*v*/*v*) formic acid in water and mobile phase B was acetonitrile. The gradient used was: 0–11 min, gradient from 25% to 95% B; 11–15 min, gradient from 95% to 25% B.

### 4.10. Statistical Analysis

All data are presented as the mean ± standard error (SE) of independent experiments and were analyzed using GraphPad Prism (GraphPad Software, San Diego, CA, USA). Comparisons between groups were performed using the Student’s *t* test. *p* < 0.05 was considered significant.

## 5. Conclusions

In conclusion, our data demonstrate that treatment of HY7015 and LT improves hair regrowth rate by upregulating hair follicle formation, and secretion of growth factors, such as VEGF and IGF-1, in animal experiments. These improvements were mediated in part by proliferation effect of HY7015 and LT. HY7015 and LT also showed cytoprotective effects by regulating SOD1 and IL-1β expressions in oxidative stress-induced HFDPC. These results suggest that this hair regrowth effect is related to the rosmarinic acid content in LT and synergistic effects of HY7015. However, our research focused on hair growth rather than detailed mechanisms. Therefore, further studies are needed to clarify the mechanisms.

## Figures and Tables

**Figure 1 molecules-27-08235-f001:**
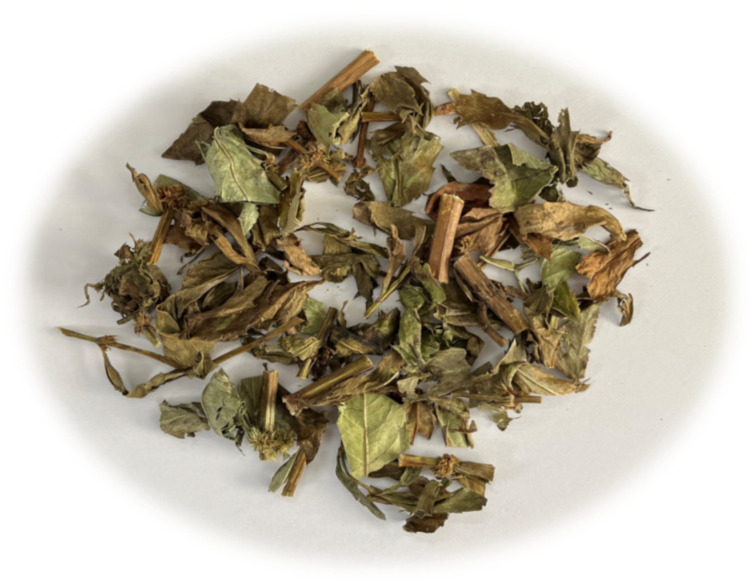
The plant studied, *Lycopus lucidus* Turcz.

**Figure 2 molecules-27-08235-f002:**
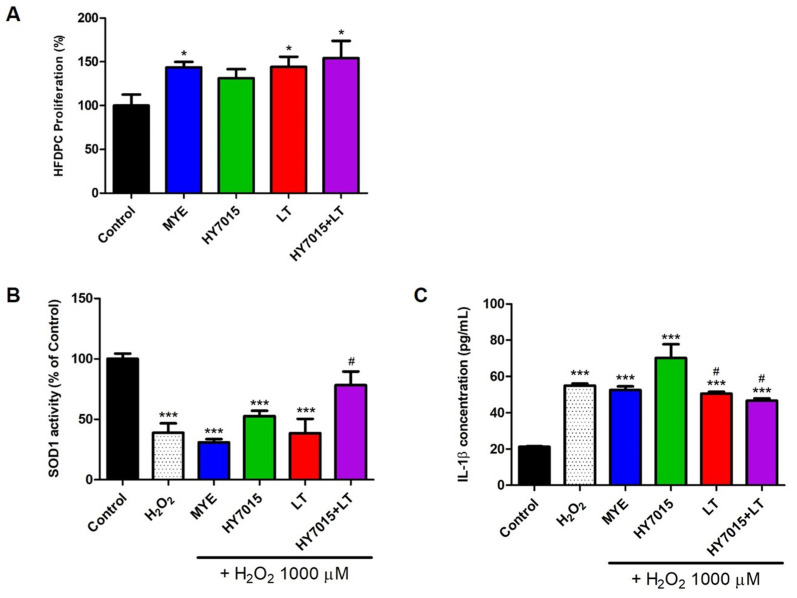
Cytoprotective effects of *Lactobacillus paracasei* HY7015 (HY7015) and/or *Lycopus lucidus* Turcz. (LT) extract in HFDPC. Effect of *Lactobacillus paracasei* HY7015 (HY7015) and/or *Lycopus lucidus* Turcz. (LT) extract on HFDPC proliferation (**A**). HFDPC were untreated (Control), or treated with medicinal yeast extract (MYE, 50 µg/mL), or with HY7015 (1 × 10^6^ CFU/mL) and/or LT (50 µg/mL) for 24 h. SOD1 activity (**B**) and IL-1β (**C**) were measured by ELISA kit. Data are presented as mean ± SE. Significant differences are indicated by * *p* < 0.05 and *** *p* < 0.001, relative to the control group. # *p* < 0.05 compared with H_2_O_2_ group. SE, standard error; MYE, medicinal yeast extract; HY7015, *Lactobacillus paracasei* HY7015; LT, *Lycopus lucidus* Turcz.; HY7015+LT, *Lactobacillus paracasei* HY7015 with *Lycopus lucidus* Turcz.

**Figure 3 molecules-27-08235-f003:**
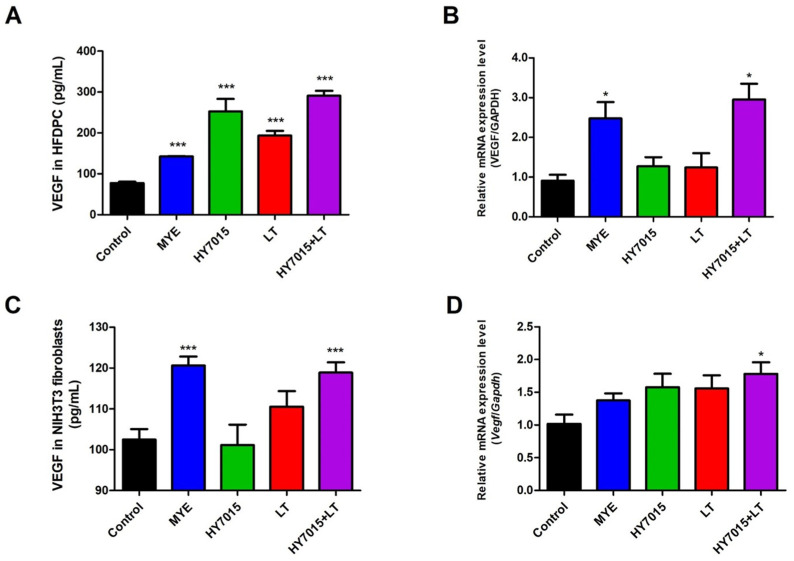
Effect of *Lactobacillus paracasei* HY7015 (HY7015) and/or *Lycopus lucidus* Turcz. (LT) extract on VEGF proteins (**A**) and mRNA (**B**) of HFPDC, VEGF proteins (**C**), and Vegf mRNA (**D**) of NIH3T3 fibroblasts. Protein levels of VEGF were measured by ELISA kit. Relative mRNA levels of VEGF were monitored by qPCR and normalized against GAPDH. Data are presented as mean ± SE. Significant differences are indicated by * *p* < 0.05 and *** *p* < 0.001, relative to the control group. SE, standard error; VEGF, vascular endothelial growth factor; MYE, medicinal yeast extract; HY7015, *Lactobacillus paracasei* HY7015; LT, *Lycopus lucidus* Turcz.; HY7015+LT, *Lactobacillus paracasei* HY7015 with *Lycopus lucidus* Turcz.

**Figure 4 molecules-27-08235-f004:**
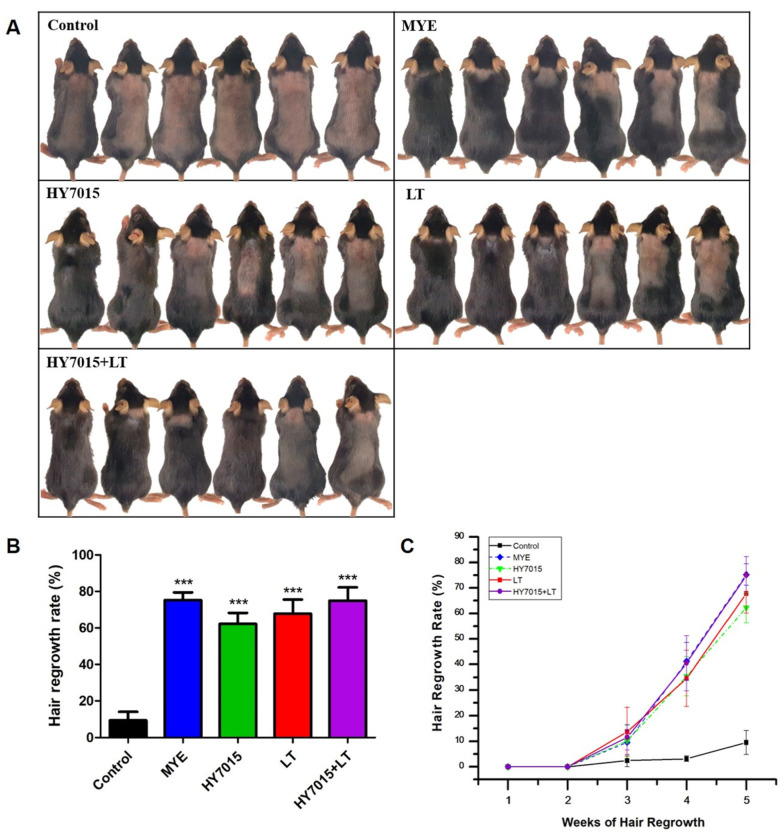
Effect of *Lactobacillus paracasei* HY7015 (HY7015) and/or *Lycopus lucidus* Turcz. (LT) extract on promoting hair regrowth in a telogenic mouse model. Shaved 7-week-old mice were fed a normal control diet (Control), or medicinal yeast extract (MYE, 200 mg/kg), or HY7015 (1 × 10^8^ CFU/day) and/or LT extract (100 mg/kg) daily for 5 weeks. (**A**) Photographs of mouse dorsal skin by groups at week 5. (**B**) Hair regrowth rates during the 5 weeks period. (**C**) Weekly assessment of hair regrowth rates in each group during the 5 weeks period. Data are presented as mean ± SE. Significant differences are indicated by *** *p* < 0.001, relative to the control group. SE, standard error; MYE, medicinal yeast extract; HY7015, *Lactobacillus paracasei* HY7015; LT, *Lycopus lucidus* Turcz.; HY7015+LT, *Lactobacillus paracasei* HY7015 with *Lycopus lucidus* Turcz.

**Figure 5 molecules-27-08235-f005:**
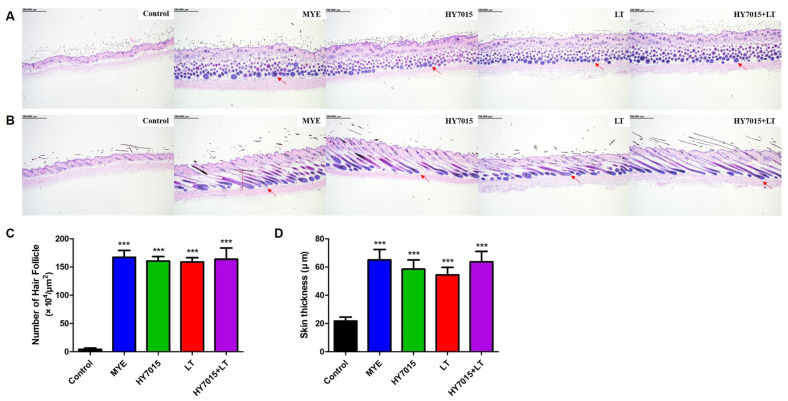
Oral administration of *Lactobacillus paracasei* HY7015 (HY7015) and/or *Lycopus lucidus* Turcz. (LT) extract promotes formation of hair follicles in telogenic mice. H&E staining histology analysis of (**A**) transverse sections. Slides were prepared by cutting the skin tissue perpendicular to the tail to observe mouse hair follicles. (**B**) Longitudinal sections for dermal layer thickness measurements of mouse dorsal skin tissues. Arrows indicate the presence of hair follicles in skin. (**C**) Numbers of hair follicles in dorsal skin layer samples (×10^4^/µm^2^). (**D**) Dermal skin thickness (μm). Data are presented as mean ± SE. Significant differences are indicated by *** *p* < 0.001, relative to the control group. SE, standard error; MYE, medicinal yeast extract; HY7015, *Lactobacillus paracasei* HY7015; LT, *Lycopus lucidus* Turcz.; HY7015+LT, *Lactobacillus paracasei* HY7015 with *Lycopus lucidus* Turcz.

**Figure 6 molecules-27-08235-f006:**
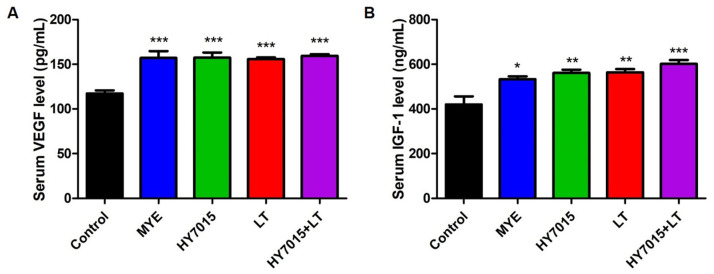
Effect of *Lactobacillus paracasei* HY7015 (HY7015) and/or *Lycopus lucidus* Turcz. (LT) extract on growth factor secretion in mouse serum. (**A**) VEGF levels in mouse serum. (**B**) IGF-1 levels in mouse serum. VEGF and IGF-1 secretions were measured by ELISA. Data are presented as mean ± SE. Significant differences are indicated by * *p* < 0.05, ** *p* < 0.01, and *** *p* < 0.001, relative to the control group. SE, standard error; MYE, medicinal yeast extract; HY7015, *Lactobacillus paracasei* HY7015; LT, *Lycopus lucidus* Turcz.; HY7015+LT, *Lactobacillus paracasei* HY7015 with *Lycopus lucidus* Turcz.; ELISA, enzyme-linked immunosorbent assay.

**Figure 7 molecules-27-08235-f007:**
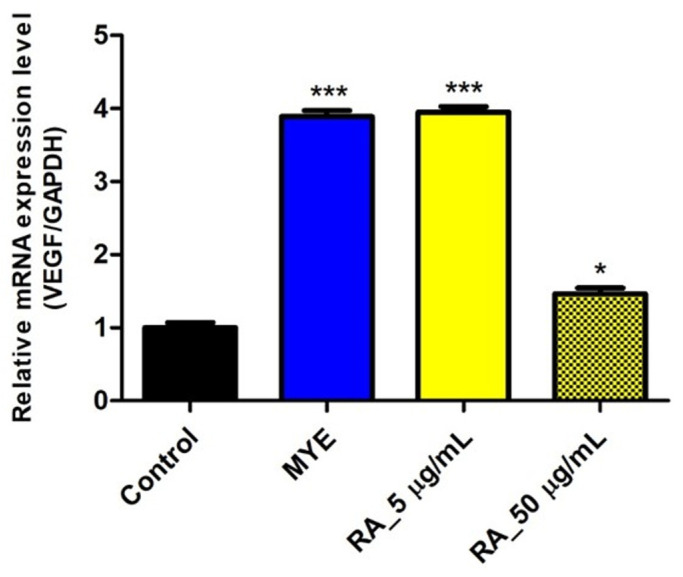
Effect of rosmarinic acid on VEGF mRNA expressions in HFDPC. Relative mRNA levels of VEGF were monitored by qPCR and normalized against GAPDH. Data are presented as mean ± SE. Significant differences are indicated by * *p* < 0.05 and *** *p* < 0.001, relative to the control group. SE, standard error; VEGF, vascular endothelial growth factor; MYE, medicinal yeast extract; RA, rosmarinic acid.

**Figure 8 molecules-27-08235-f008:**
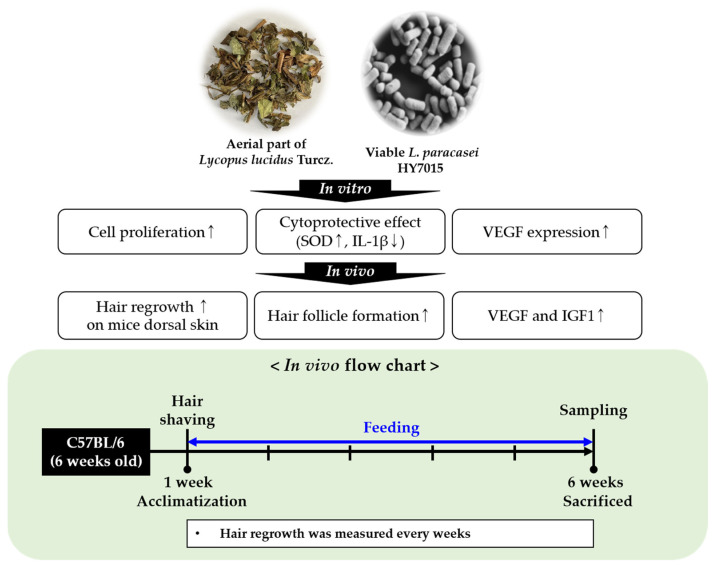
The flowchart of in vitro and in vivo.

**Table 1 molecules-27-08235-t001:** Rosmarinic acid content of *Lycopus lucidus* Turcz. extract.

Sample	Rosmarinic Acid Content (μg/mL)
Water Extract of *Lycopus lucidus* Turcz.	67.64

DAD detection at λ = 330 nm.

## Data Availability

The data presented in this study are available in the article and Appendix A. The raw data are available upon a reasonable request from the corresponding author.

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
