# Peer review of "Lactobacillus paracasei HY7015 and Lycopus lucidus Turcz. Extract Promotes Human Dermal Papilla Cell Cytoprotective Effect and Hair Regrowth Rate in C57BL/6 Mice"

_molecules, 2022, doi:10.3390/molecules27238235_

Round 1

Reviewer 1 Report

Dear authors,

Please consider the following comments to improve the content of your manuscript before publication. 

General comments:

1- Please provide the DOI number for all references cited in this manuscript. 

2- Please combine the figures to reduce the number of inserted images in the paper. Indeed, please use a colorful pattern to show your graphs. This increase the beauty of your illustrations for academic readers. 

3- Please add a flowchart to the paper to summarize the M&M section. 

4- Please check the grammatical structure of your paper. Some sentences in the introduction and results should be elaborated using active verbs. 

Specific comments:

1- Figure 5, section A: Please add the original size of this part of the figure to the supplementary files. Please note that the details of this part of figure 5 are not transparent enough; therefore, the respected authors should re-manage the sections of this figure.  

2- The introduction is too long. Please summarize it into at least three paragraphs and try to highlight the pivotal notes only. More than 40 references have been used in this section. While you revise this section, please also reduce the number of references used to prepare this section. 

3- Please add a Lycopus lucidus figure to the paper. The authors can insert its figure in the introduction section or add it to the discussion. 

4- Line 354: Please elaborate on where LT samples were collected. If the authors collected this plant from nature, providing an identification botany voucher for your samples is essential.   

5- This plant has many bioactive secondary metabolites. Previous studies have shown that different phenolics and other metabolites have been isolated from this plant. Please determine the domineering metabolites for this plant. I highly recommend the authors provide a GC/MS profile of the prepared extract to disclose the abundance of metabolites while conducting this study. 

6- Please elaborate on how oral administration of the studied bacteria can be associated with hair growth. This requires an in-depth knowledge of interactions in the gut-skin axis. Please provide more evidence for this case because the administration of probiotics requires data on their metabolic profile or interaction with body receptors. Enough evidence has not been provided for this claim. 

7- Table 1: Please provide an HPLC chromatogram for the identified compound. 

Please amend the requested comments and submit the revision file. I will keep you posted if I find further queries with your paper. Overall, it is suitable for publication. 

Author Response

Dear reviewer 1

Thank you for reviewing our manuscript for publication on molecules. We are very pleasured to have the opportunity to revise our manuscripts, and appreciate for your insights. As the reviewer noted, we agree that the gut-skin axis needs to be discussed in-depth. Regarding this point, the manuscripts were re-written with various references to help understanding. We sincerely appreciate all valuable comments and suggestions, which greatly help us improve the quality of our articles. We hope that the manuscript is now acceptable for publication in molecules and declare that authors of this work have no conflict of interests.

Sincerely,

Jung-Lyoul Lee, Ph.D

General comments:

1-Please provide the DOI number for all references cited in this manuscript.

- Answer: Thank you for your kindly comments. We have provided a DOI number or URL except for references [22].

2- Please combine the figures to reduce the number of inserted images in the paper. Indeed, please use a colorful pattern to show your graphs. This increase the beauty of your illustrations for academic readers.

- Answer: We sincerely appreciate comments, which helped us to improve the quality of the article. We changed Figure 1-2 to Figure 1. Also, we improved our article with colorful pattern.

3- Please add a flowchart to the paper to summarize the M&M section.

-Answer: Thank you for your kindly advice. We added the flowchart to the paper in M&M section (Line 390 and 391).

4- Please check the grammatical structure of your paper. Some sentences in the introduction and results should be elaborated using active verbs.

-Answer: We appreciate for pointing out our mistake. We modified some sentence, as your advice (Line 47, 55, 57, 58, 60, 61, 62, 68).

Specific comments:

1- Figure 5, section A: Please add the original size of this part of the figure to the supplementary files. Please note that the details of this part of figure 5 are not transparent enough; therefore, the respected authors should re-manage the sections of this figure.

-Answer 1: Thank you for your kindly advice. We submit the supplementary files for Figure 5, Section A (Revised Figure 4, Section A and B). As your advice, our explanation was insufficient. Thanks for giving us the opportunity to improve our article. Regarding on it, we added this paragraph in Figure 4 legend (Line 187-196);

“Oral administration of Lactobacillus paracasei HY7015 (HY7015) and/or Lycopus lucidus Turcz. (LT) extract promotes formation of hair follicles in telogenic mice. H&E staining histology analysis of (A) transverse sections. Slides were prepared by cutting the skin tissue perpendicular to the tail to observe mouse hair follicles. (B) Longitudinal sections for dermal layer thickness measurement of mouse dorsal skin tissues. Arrows indicate the presence of hair follicles in skin. (C) Numbers of hair follicles in dorsal skin layer samples (× 104/µm2). (D) Dermal skin thickness (μm). Data are presented as mean ± SE. Sig-nificant differences are indicated by *** p < 0.001, relative to the control group. SE, standard error; MYE, medicinal yeast extract; HY7015, Lactobacillus paracasei HY7015; LT, Lycopus lucidus Turcz.; HY7015+LT, Lactobacillus paracasei HY7015 with Lycopus lucidus Turcz.”

2- The introduction is too long. Please summarize it into at least three paragraphs and try to highlight the pivotal notes only. More than 40 references have been used in this section. While you revise this section, please also reduce the number of references used to prepare this section.

-Answer 2: Thank you for advice. We followed your advice, we summarized introduction and reduced the number of introduction references (Ref. [1~25]).

3- Please add a Lycopus lucidus figure to the paper. The authors can insert its figure in the introduction section or add it to the discussion.

-Answer 3: We sincerely appreciate comments, which helped us to improve the quality of the article. According to your suggestion, we have inserted the Lycopus lucidus figure into the discussion sections (Line 249).

4- Line 354: Please elaborate on where LT samples were collected. If the authors collected this plant from nature, providing an identification botany voucher for your samples is essential.  

-Answer 4: Thank you for advice. We recorded where we bought LT samples in M&M (Line 354). “100 g of dried natural plant and aerial LT samples (Humanherb, Daegu-si, Korea) were ground, mixed, ….”.

5- This plant has many bioactive secondary metabolites. Previous studies have shown that different phenolics and other metabolites have been isolated from this plant. Please determine the domineering metabolites for this plant. I highly recommend the authors provide a GC/MS profile of the prepared extract to disclose the abundance of metabolites while conducting this study.

- Answer 5: Thanks for your advice. Through previous papers, we confirmed that Lycopus lucidus Turcz. is rich in various organic acids, flavonoids, triperpenoids, and phenolic acids. Understandably, GC/MS analysis is considered necessary, but rosmarinic acid is known to be the main substance in many previous studies. In 24 samples of L. lucidus Turcz. purchased from different pharmacies in China, 78.6 – 2540 μg/g of rosmarinic acid was detected [1]. Therefore, we had to focus on rosmarinic acid. In addition, it was also confirmed that rosmarinic acid affects VEGF mRNA expression in HFDPC. Therefore, rosmarinic acid was mentioned as the domineering metabolites for this plant.

  1. Ren, Q.; Ding, L.; Sun, S. S.; Wang, H. Y.; Qu, L., Chemical identification and quality evaluation of Lycopus lucidus Turcz by UHPLC-Q-TOF-MS and HPLC-MS/MS and hierarchical clustering analysis. Chromatogr. 2017, 31, e3867, doi:10.1002/bmc.3867.

6- Please elaborate on how oral administration of the studied bacteria can be associated with hair growth. This requires an in-depth knowledge of interactions in the gut-skin axis. Please provide more evidence for this case because the administration of probiotics requires data on their metabolic profile or interaction with body receptors. Enough evidence has not been provided for this claim.

- Answer 6: We appreciate your insight in improving the quality of our articles. Depending on your advice, we think that the gut-skin axis is the part that needs more consideration. Research on the intestinal environment and skin health is being actively conducted. Bacterial metabolites like SCFAs, dopamine, and serotonin are known to directly or indirectly affect to skin health such as anti-inflammatory effects, inhibition effect of hair growth, and melatonin modulation. An association between gut bacterial imbalance and alopecia, alopecia may be affected by microorganisms that induce Th1 response, which related to the JAK/signal transducer and activator of transcription (STAT) signal pathway. These pathways can cause abnormal growth of hair follicle cells. In addition, the research showed that hair growth in 2 patients with alopecia areata who received fecal transplantation also support the gut-skin axis. In a clinical trial using strains of Lactobacillus genus, which had skin protective effects in vitro and in vivo, studies were conducted on the effects on skin health through the regulations of intestinal and immune response. Regulatory effects of Lactobacillus plantarum HY7714 on skin health by improving intestinal condition. Also, studies on the gut-brain-skin axis showed that the microbiome is regulated by the Lactobacillus genus, thereby reducing stress-induced neurodermatitis and affecting hair follicle circulation. For this reason, our study should be conducted clinical studies about the gut-skin axis. Therefore, regarding on it, we added this paragraph in discussion (Line 330-340);

“Recently studies reported that the relations between intestinal environment and skin health [1,2]. Gut microbial metabolites such as short chain fatty acids (SCFAs), dopamine and serotonin are known to directly or indirectly affect skin health through anti-inflammatory, hair follicle inhibition, and melatonin-modulating. Furthermore, genes associated with alopecia areata affect intestinal colonization of microorganisms that induce Th1 responses, which can lead to abnormal growth of hair follicle cells. In addition, the benefits of Lactobacillus genus consumption have also been studied on skin health through gut-modulating and immune responses [3,4]. HY7015 is considered to have the effects on genes related to hair growth by regulating intestinal balance and immune response system. However, further research, including clinical studies, should be conducted to support that.”

  1. O'Neill, C. A.; Monteleone, G.; McLaughlin, J. T.; Paus, R., The gut‐skin axis in health and disease: a paradigm with therapeutic implications. Bioessays 2016, 38, 1167-1176, doi:10.1002/bies.201600008.
  2. De Pessemier, B.; Grine, L.; Debaere, M.; Maes, A.; Paetzold, B.; Callewaert, C., Gut–skin axis: current knowledge of the interrelationship between microbial dysbiosis and skin conditions. Microorganisms 20219, 353, doi:10.3390/microorganisms9020353.
  3. Nam, B.; Kim, S. A.; Park, S. D.; Kim, H. J.; Kim, J. S.; Bae, C. H.; Kim, J. Y.; Nam, W.; Lee, J.L.; Sim, J. H., Regulatory effects of Lactobacillus plantarum HY7714 on skin health by improving intestinal condition. PLoS One 2020, 15, e0231268, doi:10.1371/journal.pone.0231268.
  4. Arck, P.; Handjiski, B.; Hagen, E.; Pincus, M.; Bruenahl, C.; Bienenstock, J.; Paus, R., Is there a ‘gut–brain–skin axis’?.  Dermatol. 2010, 19, 401-405, doi:10.1111/j.1600-0625.2009.01060.x.

7- Table 1: Please provide an HPLC chromatogram for the identified compound.

-Answer 7: Thank you for your kindly comments. We submit the HPLC chromatogram as a supplementary 2.

Reviewer 2 Report

The work of Lee et al., proposes a potential alternative to pharmacological treatments approved and used up to now for hair loss, which involve side effects after prolonged use. The combination of plant extracts and probiotics is innovative for the treatment of diseases, although the focus of this work has to do more with the cosmetic area, it is still relevant due to the evidence found at the in vitro and in vivo level (model animal).

The innocuousness of the treatments should have been evaluated in the murine model, and more so when the inoculation is for a long time (35 consecutive days), although the stimulating effect of hair regrowth was observed in the mouse scalp, it is necessary to evaluate the cytotoxicity of the prolonged inoculation of the extracts of Lycopus lucidus (LT), Lactobacillus paracasei HY7015 (HY7015) and the combination of both (LT+HY7015), in animal organs (liver, spleen and intestine). How did you check the cytotoxicity of the treatments (LT, HY7015 and LT+HY7015) in the murine model previously to establish the inoculation scheme?

It would be important to add a representative image of the weight of the animals during the treatment period, for example week 0, 1, 2, 3, 4 and 5, or the weight gain/loss (delta) compared to the negative control group in the same points (week 0, 1, 2, 3, 4, and 5), especially for the use of medicinal yeast extract.

I consider that they should have carried out a previous test where they evaluated the viability of HY7015 after being in contact with the LT extract (MTT), if it is within their possibilities I recommend doing it, in which they evaluate the colony-forming units of HY7015 to determine if the combined treatment does not significantly affect the viability of probiotic.

Will it have the same effect observed at work if instead of using the viable HY7015 it is used inactivated (paraprobiotic)? It has been reported that the modifications that probiotics may have after subjecting them to an inactivation process (tyndallization, heat shock), can better stimulate or modulate the immune response than when they are viable.

Although they mention it as a key cytokine in the inflammatory process (IL-1B), they could take advantage of the biological material (serum) to perform a broader cytokine profile (IFN-g, TNF-a, IL-12, IL-10 , IL-6, IL-17, MCP-1), these results can offer an overview to elucidate a possible systemic mechanism that is related to the results of the growth factors VEGF and IGF-1.

Lines 13 and 14: write in italics in vitro.

Lines 61 and 64: write in italics Lactobacillus.

Line 66: write in italics in vitro and in vivo

Line 80: write in italics Lactobacillus.

Line 116: SOD1 is missing the 1.

Figure 3A: Was there really no significant difference between the concentration of HY7015 and the control group? Review this statistical analysis.

Figure 3C: Due to the closeness of the values of the 5 groups, I recommend using a segmented graph where the range of 90 – 120 pg/mL is amplified so that the standard errors are better observed.

Line 183: the text of the results does not agree with the foot of Figure 5, in the result text there are Figure 5A, 5B and 5C, but in the foot of the figure there are 5A, 5B, 5C and 5D.

Figure 5: It remains to assign the letter "B" to the row of longitudinal section micrographs and reassign the letters C and D to match the Figure caption. Enlarge the image, the arrows indicating the follicles and the measurement bar are not observed.

Line 251: I assume that "RA" is rosmarinic acid, however it is not stated previously. Put the abbreviation on line 248.

Figure 7: Why is there a difference in the expression values of MYE VEGF/GAPDH mRNA compared to Figure 3, if the experiments are under the same conditions?

Line 266: place the reference number of the previous study (35).

Line 333: write in italics in vivo.

Line 364: write in italics in vitro.

Line 394: check the incubation times of the treatments with MYE, LT, HY7015 and LT+HY7015 in the 2 evaluations carried out (SOD1 and IL-1b), they should be 20 h + 4 h (H2O2).

Line 398: you do not mention the methodology for evaluating VEGF in mouse fibroblasts. At least mention that the same methodology of human VEGF will be used.

Line 423: Was the administration of 1x108 CFU/mL of HY7015 from a fresh culture or from a frozen one at -20ºC? If it was from a frozen one, did they check the viability after a certain time?

REFERENCES

Lines 488, 509, 510, 511, 513, 514, 516 y 518: write in italics the names of journals.

Lines 530, 535, 536, 538, 540, 542, 544, 546, 547, 550, 552, 553, 555, 557, 559, 561, 562, 565, 574, 594, 600, 607, 614 y 617: write in italics the scientific names.

Lines 33, 593 y 600: write in italics in vitro a in vivo

Lines 532, 572, 590, 592 y 610: remove the issue from the journal.

Line 558. missing journal volume.

Line 576: the volume and pages of the journal are missing.

Line 581: correct the position of the journal volume.

Author Response

Dear reviewer 2

Thank you for reviewing our manuscript for publication on molecules. We are very pleasured to have the opportunity to revise our manuscripts, and appreciate for your insights. As the reviewer noted, we agree that innocuousness of the treatments and paraprobiotics needs to be discussed. Regarding this point, the manuscripts were rewritten with various references to help understanding. We sincerely appreciate all valuable comments and suggestions, which greatly help us improve the quality of our articles. We hope that the manuscript is now acceptable for publication in molecules and declare that authors of this work have no conflict of interests.

Sincerely,

Jung-Lyoul Lee, Ph.D

The innocuousness of the treatments should have been evaluated in the murine model, and more so when the inoculation is for a long time (35 consecutive days), although the stimulating effect of hair regrowth was observed in the mouse scalp, it is necessary to evaluate the cytotoxicity of the prolonged inoculation of the extracts of Lycopus lucidus (LT), Lactobacillus paracasei HY7015 (HY7015) and the combination of both (LT+HY7015), in animal organs (liver, spleen and intestine). How did you check the cytotoxicity of the treatments (LT, HY7015 and LT+HY7015) in the murine model previously to establish the inoculation scheme?

-Answer 1: Thank you for your kindly advice. We confirmed the cytotoxicity of the treatments (LT, HY7015, LT+HY7015) in vitro, however did not perform the cytotoxicity of the treatments in the murine model. Through many papers, two substances have been found to be safe. There are still many questions about whether probiotics is safe to consume, but its safety has been proven in various clinical trials and experiments using murine models in lactic acid bacteria, including Lactobacillus paracasei [1,2]. LT is a plant that has long been used medicinally in Asia, and previous studies in mast cell and murine models has supported the safety of ingestion [3,4]. Cytotoxicity assessments between Lactobacillus paracasei and LT were difficult to find in the prior study. Instead, we evaluated viable cell counts for 2 months to confirm whether LT affected the growth of HY7015 or not (we hope “Answer 3” helps reviewers understand.). Based on your comments, we thought that various toxicity evaluations should be performed in animal organs with HY7015+LT. In addition, it is considered necessary to proceed with Next-Generation Sequencing (NGS) analysis of HY7015 to check the antibiotic resistance, production of toxin, etc.

  1. Ezendam, J.; van Loveren, H., Probiotics: immunomodulation and evaluation of safety and efficacy.  Rev.2006, 64, 1-14, doi:10.1111/j.1753-4887.2006.tb00168.x.
  2. Martín, R.; Chain, F.; Miquel, S.; Motta, J. P.; Vergnolle, N.; Sokol, H.; Langella, P., Using murine colitis models to analyze probiotics–host interactions. FEMS Microbiol. Rev.2017, 41, S49-S70, doi:10.1093/femsre/fux035.
  3. Shin, T. Y.; Kim, S. H.; Suk, K.; Ha, J. H.; Kim, I.; Lee, M. G.; Kim, H. M., Anti-allergic effects of Lycopus lucidus on mast cell-mediated allergy model. Appl. Pharmacol. 2005, 209, 255-262, doi:10.1016/j.taap.2005.04.011.
  4. Lu, Y. H.; Tian, C. R.; Gao, C. Y.; Wang, W. J.; Yang, W. Y.; Kong, X.; Liu, Z. Z., Protective effect of free phenolics from Lycopus lucidus root on carbon tetrachloride-induced liver injury in vivo and in vitro. Food Nutr. Res. 201862, doi:10.29219/fnr.v62.1398.

It would be important to add a representative image of the weight of the animals during the treatment period, for example week 0, 1, 2, 3, 4 and 5, or the weight gain/loss (delta) compared to the negative control group in the same points (week 0, 1, 2, 3, 4, and 5), especially for the use of medicinal yeast extract.

-Answer 2: Thank you for your advice which can enhance the quality of our paper. I’m afraid that we can only provide photographs of mice dorsal skin and hair regrowth rate (%) in Figure 3. In Figure 3C, the degree of hair regrowth and non-regrowth were measured as a ratio every week. Weight of the animals during treatment period was not measured during the experiment. Because the degree of hair growth is different for each mouse, it was determined that this may affect the weight of the mouse. It would be best if we had performed an animal study that considered various factors such as metabolism, obesity, and immunity. However, we could not take this into consideration as we only focused on hair regrowth. As your advice, further research should consider this aspect.

I consider that they should have carried out a previous test where they evaluated the viability of HY7015 after being in contact with the LT extract (MTT), if it is within their possibilities I recommend doing it, in which they evaluate the colony-forming units of HY7015 to determine if the combined treatment does not significantly affect the viability of probiotic.

-Answer 3: Thank you for your comments. Through our studies, the LT extract is unlikely to affect the probiotics properties. Because the mass-produced HY7015 freeze-dried product and HY7015 (10% of freeze-dried HY7015) + LT (1g) complex showed the same patterns as HY7015 only during storage;

> Baseline: HY7015 7.70 X 1010 CFU/mL and HY7015+LT 6.25 X 109 CFU/mL,

> Week-4: HY7015 4.05 X 1010 CFU/mL and HY7015+LT 3.10 X 109 CFU/mL,

> Week-8: HY7015 3.15 X 1010 CFU/mL and HY7015+LT 2.26 X 109 CFU/mL.

Will it have the same effect observed at work if instead of using the viable HY7015 it is used inactivated (paraprobiotic)? It has been reported that the modifications that probiotics may have after subjecting them to an inactivation process (tyndallization, heat shock), can better stimulate or modulate the immune response than when they are viable.

-Answer 4: Thank you for the insight on this. We appreciate that your comments gave us second thought to this. We proceeded with the idea by using live probiotics isolated from the traditional fermented liquor could be the basis for various research such as gut-skin axis. We thought that HY7015 would have a hair health effect by improving immunity or hair follicle inflammation through microbiome regulation. We missed the consideration of paraprobiotics to focus on live probiotics, but it would be better if an experiment considering the paraprobiotics that affects immune response or hair growth.

Although they mention it as a key cytokine in the inflammatory process (IL-1B), they could take advantage of the biological material (serum) to perform a broader cytokine profile (IFN-g, TNF-a, IL-12, IL-10 , IL-6, IL-17, MCP-1), these results can offer an overview to elucidate a possible systemic mechanism that is related to the results of the growth factors VEGF and IGF-1.

-Answer 5: We sincerely appreciate your insight. We used IL-1β to confirm its cytoprotective effects in vitro prior to animal experiments. As you advice, it would have been better if the cytokines affecting VEGF and IGF-1 were measured in serum. Because some studies have demonstrated the relationship between inflammatory responses and hair loss. Other studies have shown that when hair formation is promoted, TNF-α and IL-1β protein levels significantly decreased, while IL-4 and IL-13 increased [1]. Further studies are needed to confirm the relationship between hair growth and pro- and/or anti-inflammatory cytokines in serum.

  1. Lee, T. K.; Kim, B.; Kim, D. W.; Ahn, J. H.; Sim, H.; Lee, J. C.; Won, M. H., Effects of decursin and Angelica gigas nakai root extract on hair growth in mouse dorsal skin via regulating inflammatory cytokines. Molecules2020, 25, 3697, doi:10.3390/molecules25163697.

Lines 13 and 14: write in italics in vitro.

-Answer 6: We appreciate for pointing out our mistake. We modified them according to the comments (Line 14-15).

Lines 61 and 64: write in italics Lactobacillus.

-Answer 7: We appreciate for pointing out our mistake. We modified them according to the comments (Line 42 and 44).

Line 66: write in italics in vitro and in vivo.

-Answer 8: We appreciate for pointing out our mistake. We modified them according to the comments (Line 46 and 47).

Line 80: write in italics Lactobacillus.

-Answer 9: We appreciate for pointing out our mistake. We modified them according to the comments (Line 60).

Line 116: SOD1 is missing the 1.

-Answer 10: We appreciate for pointing out our mistake. We modified them according to the comments (Line 78, 80, 93).

Figure 3A: Was there really no significant difference between the concentration of HY7015 and the control group? Review this statistical analysis.

-Answer 11: We appreciate for pointing out our mistake. As review this statistical analysis, it was confirmed that it was set incorrectly. And all statistical analyzed described in the article were rechecked, but all were the same except for Figure 3A (revised Figure 2A). Regarding on it, we modified the sentence (Line: 101-103).

Figure 3C: Due to the closeness of the values of the 5 groups, I recommend using a segmented graph where the range of 90 – 120 pg/mL is amplified so that the standard errors are better observed.

-Answer 12: Thank you for your kindly comments. As you suggested, the range has been modified (Line 120, Figure 2C).

Line 183: the text of the results does not agree with the foot of Figure 5, in the result text there are Figure 5A, 5B and 5C, but in the foot of the figure there are 5A, 5B, 5C and 5D.

-Answer 13: We appreciate for pointing out our mistake. We modified Figure (Line 186, revised Figure 4) as your advice.

Figure 5: It remains to assign the letter "B" to the row of longitudinal section micrographs and reassign the letters C and D to match the Figure caption. Enlarge the image, the arrows indicating the follicles and the measurement bar are not observed.

-Answer 14: Thank you for your kindly advice. We submit the supplementary files for Figure 5, Section A (Revised Figure 4, Section A and B).

Line 251: I assume that "RA" is rosmarinic acid, however it is not stated previously. Put the abbreviation on line 248.

-Answer 15: We appreciate for pointing out our mistake. We modified them according to the comments (Line 226).

Figure 7: Why is there a difference in the expression values of MYE VEGF/GAPDH mRNA compared to Figure 3, if the experiments are under the same conditions?

-Answer 16: Thanks for letting us know what we missed. We confirmed similar VEGF levels when we repeated the VEGF expression level in HFDPC under same conditions. However, in the case of the recent experiment, Figure 6, the stabilization step of the activated HFDPC was lacking.

Line 266: place the reference number of the previous study (35).

-Answer 17: We appreciate for pointing out our mistake. We modified them according to the comments. (Line 244)

Line 333: write in italics in vivo.

-Answer 18: We appreciate for pointing out our mistake. We modified them according to the comments. (Line 314)

Line 364: write in italics in vitro.

-Answer 19: We appreciate for pointing out our mistake. We modified them according to the comments. (Line 363)

Line 394: check the incubation times of the treatments with MYE, LT, HY7015 and LT+HY7015 in the 2 evaluations carried out (SOD1 and IL-1b), they should be 20 h + 4 h (H2O2).

-Answer 20: We appreciate for pointing out our mistake. We modified them according to the comments. We modified the sentence in M&M (Line 404-406);

 “For H2O2 treatment, the samples were treated for 20 h and then treated with 1000 μM H2O2 for 4 h, and then the supernatant was collected and protein was measured. To determine VEGF contents in NIH3T3 fibroblasts, the Mouse VEGF (MMV00) Quantikine ELISA kits (R&D Systems, Minneapolis, MN, USA) were used according to the manufacturer’s instructions. NIH3T3 cells treated with samples for 24h, and then the supernatant was collected and protein was measured.”

Line 398: you do not mention the methodology for evaluating VEGF in mouse fibroblasts. At least mention that the same methodology of human VEGF will be used.

-Answer 21: Thank you for pointing out our mistake. We modified them according to the comments. (Line 408 and 409)

Line 423: Was the administration of 1x108 CFU/mL of HY7015 from a fresh culture or from a frozen one at -20ºC? If it was from a frozen one, did they check the viability after a certain time?

-Answer 22: Thank you for pointing out our mistake. We should have added the description of HY7015 used in animal experiments. Fresh cultured HY7015 was mass-cultured and freeze-dried for in vivo assay. After freeze-drying (FD), the number of viable cells of HY7015 were measured. Then, 1 x 108 CFU/day was provided to mice. Regarding on it, we added this paragraph in M&M 4.1. Sample preparation (Line 364 and 365):

“For in vivo assay, fresh cultured HY7015 was prepared by freeze-drying and added to feed.”

Also, we modified the sentence of Cell Culture section (Line 381 and 389) for adding:

“HY7015 (1 x 106 CFU/mL)”

REFERENCES

Lines 488, 509, 510, 511, 513, 514, 516 y 518: write in italics the names of journals.

-Answer 23: Thank you for pointing out our mistake. We modified them according to the comments. (Line 488→494, 509→574, 510-511→575-576, 513,514,516→remove and 518→500)

Lines 530, 535, 536, 538, 540, 542, 544, 546, 547, 550, 552, 553, 555, 557, 559, 561, 562, 565, 574, 594, 600, 607, 614 y 617: write in italics the scientific names.

-Answer 24: We appreciate for pointing out our mistake. We modified them according to the comments. (Line 530→507, 535-536→513-514, 538→516, 540→518, 542→520, 544→522, 546→525, 547→527, 550→531, 552→533, 553→535, 555→537, 557→539, 559→541-542, 561-562→543-544, 565→547, 574→558, 594→591, 600→597, 607→604-605, 614→612, and 617→628).

Lines 33, 593 y 600: write in italics in vitro a in vivo.

-Answer 25: Thank you for pointing out our mistake. We modified them according to the comments. (Line 46-47, 590 and 597)

Lines 532, 572, 590, 592 y 610: remove the issue from the journal.

-Answer 26: Thank you for pointing out our mistake. We modified them according to the comments. (Line 532→509, 572→555, 590→585, 592→588 and 610→607)

Line 558. missing journal volume.

-Answer 27: We sincerely appreciate your attention to details. We reflected all the contents based on that comments (Line 540).

Line 576: the volume and pages of the journal are missing.

-Answer 28: We sincerely appreciate your attention to details. We reflected all the contents based on that comments (Line 560-561).

Line 581: correct the position of the journal volume.

-Answer 29: We sincerely appreciate your attention to details. We reflected all the contents based on that comments (Line 573).

We appreciate all of your kindly comments. We hope our answers will help reviewers better understand, and the manuscript is now available for publication in molecules.

Reviewer 3 Report

The authors of the publication entitled "Lactobacillus paracasei HY7015 and Lycopus lucidus Turcz. 2 Extract Promotes Human Dermal Papilla Cell Cytoprotective Effect and Hair Regrowth Rate in C57BL/6 Mice" present the effect of probiotic bacteria together with plant polyphenols on the hair of mice. The analyzed topic is very interesting, the research includes the effects of probiotic bacteria, and this is part of a very current research trend. The research is well planned. However, I have some remarks and comments which I have given below.

- authors mention in the introduction that „…plant extracts were shown to enhance the viability and health effects of probiotic Lactobacillus strains” (line 79). Has the effect of tested extract on the viability of Lactobacillus paracasei been confirmed?

 - Although the source of isolation of the strain was given in the "Material and methods", there is no information about the method of its identification

- which guided the selection from among all components of the plant extract of rosmarinic acid. Was it the dominant component, was it the strongest effect on hair growth, or maybe there was another reason. Why weren't experiments with this acid performed with the whole extract only?

 - MYE was used as a positive control in the study, but it was not stated in the text why this compound was used in the study, e.g., in the „Material and methods” section.

- Since the authors emphasized that the role of polyphenolic compounds and probiotic bacteria on the condition of the hair, it is worth noting what differentiates these papers from those and what these results bring to the already existing knowledge. It is not mentioned very much in the discussion.

- there are a few editorial errors in the text, e.g., the names of microorganisms are not written in italics and abbreviated phrases, e.g., „2.1.2. Antioxidants and Anti-inflammatory Effects on HFDPC” effect what factor? Line 360 „HY7015 was isolated from Makgeolli…” should be Lactobacillus paracasei HY7015. Authors should review the manuscript in this regard

- last year, the authors published papers under the title „Lactobacillus paracasei HY7015 Promotes Hair Growth in a Telogenic Mouse Model”, I do not have access to the full version of this work, and I do not know if the results of these papers do not coincide to any degree. I will ask the authors to comment on this.

Author Response

Dear reviewer 3

Thank you for reviewing our manuscript for publication on molecules. We are very pleasured to have the opportunity to revise our manuscripts, and appreciate for your insights. As the reviewer noted, we agree that the role of polyphenolic compounds and probiotics on hair growth needs to be discussed. Regarding this point, the manuscripts were rewritten with various references to help understanding. We sincerely appreciate all valuable comments and suggestions, which greatly help us improve the quality of our articles. We hope that the manuscript is now acceptable for publication in molecules and declare that authors of this work have no conflict of interests.

Sincerely,

Jung-Lyoul Lee, Ph.D

- authors mention in the introduction that „…plant extracts were shown to enhance the viability and health effects of probiotic Lactobacillus strains” (line 79). Has the effect of tested extract on the viability of Lactobacillus paracasei been confirmed?

- Answer: Thank you for your comments. Through our studies, the LT extract is unlikely to affect the probiotics properties. Because the mass-produced HY7015 freeze-dried product and HY7015 (10% of freeze-dried HY7015) + LT (1g) complex showed the same patterns as HY7015 only during storage:

> Baseline; HY7015 7.70 X 1010 CFU/mL and HY7015+LT 6.25 X 109 CFU/mL,

> Week-4; HY7015 4.05 X 1010 CFU/mL and HY7015+LT 3.10 X 109 CFU/mL,

> Week-8; HY7015 3.15 X 1010 CFU/mL and HY7015+LT 2.26 X 109 CFU/mL.

 - Although the source of isolation of the strain was given in the "Material and methods", there is no information about the method of its identification

-Answer: We sincerely appreciate comments, which helped us to improve the quality of the article. Regarding on it, we added this paragraph in M&M sections (Line 365-370):

“Strains were identified through 16S rRNA sequencing (Macrogen, Inc., Seoul, Korea) using universal rRNA gene primers (785F and 907R). 16S rRNA sequencing results were compared to the Genbank database via the Basic Local Alignment Search Tool (BLAST) of the National Center for Biotechnology Institute (NCBI). The sequenced strain was named L. paracasei HY7015 and deposited with KCTC (KCTC, Jeonge-up, Jeollabuk-do, Korea) with accession number KCTC14004BP.”

- which guided the selection from among all components of the plant extract of rosmarinic acid. Was it the dominant component, was it the strongest effect on hair growth, or maybe there was another reason. Why weren't experiments with this acid performed with the whole extract only?

-Answer: Thank you for your kindly advice. Rosmarinic acid is known to be the main substance in many previous studies. In 24 samples of L. lucidus Turcz. purchased from different pharmacies in China, 78.6 – 2540 μg/g of rosmarinic acid was detected [1]. Therefore, we had to focus on rosmarinic acid. In addition, it was also confirmed that rosmarinic acid affects VEGF mRNA expression in HFDPC. Therefore, rosmarinic acid was mentioned as the domineering metabolites for this plant. Following your advice, further research should consider the exploration of the various components from LT.

  1. Ren, Q.; Ding, L.; Sun, S. S.; Wang, H. Y.; Qu, L., Chemical identification and quality evaluation of Lycopus lucidus Turcz by UHPLC-Q-TOF-MS and HPLC-MS/MS and hierarchical clustering analysis. Biomed. Chromatogr. 2017, 31, e3867, doi:10.1002/bmc.3867.

 - MYE was used as a positive control in the study, but it was not stated in the text why this compound was used in the study, e.g., in the „Material and methods” section.

- Answer: Thank you for your advice on this. As your advice, our explanation was insufficient. Thanks for giving us the opportunity to improve our article. MYE, medicinal yeast, was used as it is edible drug, Also, it is a scientifically analyzed standardized ingredient that has been clinically proven to be effective in hair growth. Regarding on it, we added a paragraph in M&M section. (Line 370-373):

“Minoxyl-S (Hyundai Pharm. Co., Ltd., Seoul, Korea) was purchased and used as positive control. The main component of Minoxyl-S is medicinal yeast extract (MYE), which supplies essential nutrients for hair growth.”

- Since the authors emphasized that the role of polyphenolic compounds and probiotic bacteria on the condition of the hair, it is worth noting what differentiates these papers from those and what these results bring to the already existing knowledge. It is not mentioned very much in the discussion.

-Answer: We appreciate your insight in improving the quality of our articles. As your advice, we emphasize the role of polyphenolic compounds from LT and probiotics in hair health. However, there is insufficient explained in the discussion section. We were looking for substances that could achieve hair health with less side effects. In addition, we would like to further develop plant’s study through research on plants, which do not have sufficient functionality. Various factors influence the growth and inhibition of microorganisms [1]. Previous research has shown that fermentation with herbs, which have highly polyphenols, affects the growth and inhibition of various microorganisms including probiotics [2]. However, this has not yet been fully discussed and its effect on admixture. Furthermore, our study needs to explore how different polyphenols contained in LT affect probiotics. Regarding on it, we modified the discussion section. (Line 340-347):

"In addition, diverse factors influence the growth and inhibition of microorganisms. In particular, probiotics are known to influence growth during fermentation by carbon, nitrogen, trace elements as well as polyphenols [1]. Also, recent studies have shown that fermentation for 24 hours with polyphenols from various plants regulates the growth and inhibition of microorganisms [2]. Various polyphenolic compounds related to LT have been reported, but research on the relationship between LT and probiotics has not progressed. Therefore, our paper provides a potential possibility that LT affects HY7015 growth.”

  1. Hervert-Hernández, D.; Goñi, I., Dietary polyphenols and human gut microbiota: a review. Food Rev. Int.2011, 27, 154-169, doi:10.1080/87559129.2010.535233.
  2. Milutinović, M.; Dimitrijević-Branković, S.; Rajilić-Stojanović, M., Plant extracts rich in polyphenols as potent modulators in the growth of probiotic and pathogenic intestinal microorganisms.  Nutr.2021, 8, doi:10.3389/fnut.2021.688843.

- there are a few editorial errors in the text, e.g., the names of microorganisms are not written in italics and abbreviated phrases, e.g., „2.1.2. Antioxidants and Anti-inflammatory Effects on HFDPC” effect what factor? Line 360 „HY7015 was isolated from Makgeolli…” should be Lactobacillus paracasei HY7015. Authors should review the manuscript in this regard

-Answer: We appreciate for pointing out our mistake. We modified them according to the comments (Line 14-15, 42, 44, 46-47, 60, 314 and 363).

- last year, the authors published papers under the title „Lactobacillus paracasei HY7015 Promotes Hair Growth in a Telogenic Mouse Model”, I do not have access to the full version of this work, and I do not know if the results of these papers do not coincide to any degree. I will ask the authors to comment on this.

-Answer: Thank you for your comments. We apologies for the difficulty in accessing the paper “Lactobacillus paracasei HY7015 Promotes Hair Growth in a Telogenic Mouse Model (doi: 10.1089/jmf.2020.4860)”. As it is a paid journal in Korea, it is judged that there is difficulty in accessing it. This paper is; HY7015 can stimulate hair growth by promoting growth factor secretion and dermal papilla cell proliferation, both in vitro and in mouse experiments. This study highlights the potential for further expansion of lactic acid bacteria functional applications and could stimulate further research into foods that have no side effects and can safely stimulate hair growth. We hope our answers will help reviewers better understand.

Round 2

Reviewer 1 Report

Dear authors

Thanks for your answer. I have no further comments, and my recommendation for this paper is [acceptance in the present form]. I hope all the best to the respected authors. 

Reviewer 3 Report

Dear Authors,

thank you for taking my comments into account and making changes to the text. I still have doubts about the published article on a similar topic using the strain tested in this work, but if the Editor had the opportunity to read the full version of that paper and there are no comments, then I accept  manuscript in this form.